# Anxiety and Depression Associated with Anal Sexual Practices among HIV-Negative Men Who Have Sex with Men in Western China

**DOI:** 10.3390/ijerph17020464

**Published:** 2020-01-10

**Authors:** Jiaxiu Liu, Xiaoni Zhong, Zhuo Lu, Bin Peng, Yan Zhang, Hao Liang, Jianghong Dai, Juying Zhang, Ailong Huang

**Affiliations:** 1School of Public Health and Management, Chongqing Medical University, Chongqing 400016, China; 2Department of Epidemiology and Medical Statistics, School of Public Health, Guangxi Medical University, Nanning 520021, China; 3Department of Epidemiology and Health Statistics, School of Public Health, Xinjiang Medical University, Xinjiang 830011, China; 4Department of Epidemiology and Medical Statistics, School of Public Health, Sichuan University, Chengdu 610041, China; 5Key Laboratory of Molecular Biology on Infectious Diseases, Ministry of Education, Chongqing Medical University, Chongqing 400016, China

**Keywords:** anxiety, depression, HIV-negative MSM, anal sex role, influencing factors, western China

## Abstract

This study aimed to explore the prevalence and influencing factors of anxiety and depression among human immunodeficiency virus negative (HIV-negative) men who have sex with men (MSM) based on anal sex roles, so as to provide a scientific basis for the management of mental health conditions. Data were obtained from the baseline in a cohort study with a two-year follow-up period in western China. The Self-Rating Anxiety Scale and Center for Epidemiological Studies Depression Scale were used to assess anxiety and depression symptoms, respectively. The prevalence of anxiety and depression in 1771 MSM was 26.03% and 37.83%. Among them, 182 anal sex role “0” MSM who only had receptive anal sex with men reported the highest prevalence of anxiety and depression (31.32% and 46.15%), 467 anal sex role “1” MSM who only had insertive anal sex with men (22.27% and 32.76%), and 1122 anal sex role “0.5” MSM who engaged equally in both insertive and receptive anal sex intercourse with men (26.74% and 38.59%), respectively. Logistic regression analysis revealed that the influencing factors of anxiety include anal sex role “0”/”0.5”, low educational level, female sexual partners, frequently searching partners on the Internet, sexually transmitted diseases (STD) diagnosed by doctors, and no HIV counseling. Anal sex role “0”/”0.5”, rural area, casual male partners, female partners, STD diagnosed by doctors, frequently searching partners on the Internet, no HIV counseling, no condom use, and daily alcohol use were found to be associated with depression. In conclusion, early identification and intervention of anxiety and depression symptoms in HIV-negative MSM should be carried out, especially for anal sex role “0” MSM. Furthermore, key intervention and psychological counseling should be taken into consideration for MSM with a low education level, high-risk behaviors, and high-risk perceptions.

## 1. Introduction

In China, HIV (human immunodeficiency virus) prevalence has declined in recent years, while men who have sex with men (MSM) have become the most important high-risk group for new HIV infection [1,2,3]. The Joint UN Programme on HIV/AIDS (acquired immunodeficiency syndrome) report [4] also pointed out that MSM are 19 times more likely to be infected with HIV than the general population, which also brings a lot of mental health problems. Moreover, MSM were found to be three times more likely to suffer from mental disorders, such as anxiety and depression, than heterosexuals [5].

Globally, the social acceptance of MSM is not high in most countries and regions [6,7], and there is widespread prejudice or even discrimination against MSM, which leads to a certain degree of mental health problems in MSM. In previous studies, the prevalence of anxiety in the general population was less than 10% [8], and that of depression was between 5% and 12% [9] while the prevalence of anxiety in MSM was much higher than that of the general population [8,10,11]. The prevalence of anxiety and depression in an MSM population in Canada [12] and the United States [13] ranged from 26.4% to 44% and 39.3% to 80%, respectively, and in China [14,15,16], these figures were about 21.25% to 24.0% and 33.2% to 45.4%, respectively, suggesting that mental health in MSM could not be ignored.

Compared with traditional heterosexual behavior, anal sex and oral sex are mainly sexual behaviors of MSM, especially anal sex behaviors, which are generally expressed as “1” and “0” for insertive and receptive anal sex, respectively [17,18]. As is well-known, because of the special tissue structure of the anus, the risk of sexually transmitted diseases (STD) during anal sex is higher than that of oral sex [19,20]. Therefore, this study focused on the anxiety and depression symptoms differentiated by anal sex roles among MSM. Studies [21] have shown that MSM playing the anal sex role “1” generally have male characteristics and take an active role while MSM playing the anal sex role “0” tend to be feminine and take a passive and submissive role in male sexual intercourse. In addition, there are quite a number of MSM playing both the “1” and “0” roles, generally expressed as the anal sex role “0.5”. Therefore, there are considerable sexual role conflicts in MSM [22,23]. Different anal sex roles tend to form different sexual psychology, and the conflicts of sex roles will lead to different mental disorders, especially anxiety and depression, which are the most common symptoms [24]. However, research on anxiety and depression in relation to different anal sex roles in MSM in China are limited.

A growing number of reports [11,25] have revealed that social demographic characteristics, sexual orientation, unprotected sexual behavior, commercial sexual behavior, stigma effect, and so on are related to anxiety and depression symptoms in MSM. Moreover, Parker [26] and Fendrich [27] also suggested that alcohol use and substance use are associated with anxiety and depression. In China, due to the deep-rooted traditional culture, the requirement of society for men is to have a son to carry on his family name. As a result, many MSM have to hide their homosexual orientation through marriage, and unmarried MSM also have to perform homosexual activities underground. In addition, there are studies [28,29] that show that mental disorders of HIV-negative MSM may increase the risk of HIV infection, but research in China and other countries has always focused on the psychological condition of HIV-positive MSM [30,31,32]. As previous studies [17,33] showed, MSM in different anal sex role groups differed significantly and the risk of HIV infection was higher in the anal sex role “0” group and “0.5” group. Therefore, it is necessary to distinguish anal sex roles when exploring the influencing factors of anxiety and depression in HIV-negative MSM.

It is worth mentioning that western China (especially Chongqing and Sichuan, known as the “gay capital” in China) is referred to as a high gathering place of MSM and is the focus area of an HIV epidemic while few studies have been conducted on mental health differentiated by anal sex roles in MSM. Therefore, based on the anal sex role of MSM, this study explored the prevalence and influencing factors of anxiety and depression of HIV-negative MSM in four provinces (Chongqing, Xinjiang, Sichuan, and Guangxi) of western China for the first time, which has certain innovative and practical significance for a better understanding of the sexual culture and psychology of MSM in western China. Furthermore, it also provides a basis for formulating prevention and intervention policies and guidelines for the mental health of HIV-negative MSM in China, especially in western China.

## 2. Materials and Methods

### 2.1. Study Procedures

The data of our study were collected from a prospective cohort study “An open, randomized, multi-center, parallel-controlled clinical intervention trial conducted in western China on MSM taking antiviral drugs to prevent new HIV infections”, which was initiated by the Ministry of Science and Technology of China. From April 2013 to March 2015, a total of 2422 participants were recruited using non-probability sampling (including non-governmental organizations (NGOs), peer introduction, core members “snowball”, AIDS voluntary counseling and testing (VCT) clinics, network, etc.) in four research sites (Chongqing, Guangxi, Xinjiang, and Sichuan Province). In this study, 1771 MSM met the inclusion and exclusion criteria and completed the baseline survey and anxiety and depression scales.

Inclusion criteria: (a) Signing informed consent; (b) age ≥18 years old and ≤65 years old; (c) HIV negative; (d) average sex activity once every two weeks; and (e) at least one or more male sex couples per month before the trial.

Exclusion criteria: (a) HIV antibody positive; (b) HBsAg (hepatitis B surface antigen) or anti-HBc (hepatitis B core antibody) positive; (c) missing values in male sexual partners in the past 6 months; (d) missing values in the anal sex role; and (e) incomplete anxiety or depression scale.

### 2.2. Study Content and Measurements

#### 2.2.1. Assessment of Anxiety

Anxiety was measured using the Self-Rating Anxiety Scale (SAS) compiled by Zung in 1971 [34], having been widely used in clinical trials and psychological evaluation studies. The scale consists of 20 items, including 15 items in the forward direction and 5 items in the reverse direction. The frequency of symptoms was assessed by a four-grade score (1 = no or very little time, 2 = a few times, 3 = quite a lot of times, 4 = most or all the time). The scores for the 20 items were aggregated into a raw score, and a standard score was obtained by multiplying the raw score by 1.25 and then taking the integer. A standard score equal to or greater than 50 indicated an anxiety symptom and a higher score indicated greater levels of anxiety symptoms. The standardized Cronbach’s α in this study was 0.8207.

#### 2.2.2. Assessment of Depression

Depression was assessed with the Center for Epidemiological Studies Depression Scale (CES-D) [35]. The scale consists of 20 items, of which 16 items were positive and 4 items were negative. According to the frequency of symptoms in the last week, the respondents scored, by themselves, from 0 (occasional or no) to 3 (most of the time or persistence). With a total score of 60 points, 16 points or above were identified as having depression symptoms. The standardized Cronbach’s α in this study was 0.8192.

#### 2.2.3. Anal Sex Role Factors

All participants were requested to report “what is your usual sex position during anal sexual activities with men?” at baseline. In our study, MSM were divided into three anal sex role subgroups: (a) Anal sex role “1” (insertive anal sex) referred to the way that MSM only had insertive anal sex with men; (b) anal sex role “0” (receptive anal sex) meant that MSM only played a receptive anal sex role with male sex partners; and (c) anal sex role “0.5” (versatile anal sex) indicated men who engaged equally in both insertive and receptive anal sex intercourse with men, also expressed as a versatile anal sex role.

#### 2.2.4. Social Demographic Factors

Social demographic factors mainly included age, urban or rural areas, ethnic groups, educational level, employment status, marital status, and personal monthly income.

#### 2.2.5. Behavior-Related Factors

Behavior-related factors mainly included sexual behaviors and substance use behaviors. Sexual behaviors of MSM in this study included the number of regular male sexual partners in the past 6 months, the number of casual male sexual partners in the past 6 months, the number of female sexual partners in the past 6 months, and the frequency of seeking sexual partners on the Internet in the past 6 months (often, sometimes or occasionally, never), whether commercial sexual service has occurred in the past 6 months, and diagnosed with an STD in the past six months. Substance use included the frequency of alcohol use in the last month (daily, occasionally, never), and the use of illicit drugs.

#### 2.2.6. Risk Perception-Related Factors

In this study, the definition of risk perception was to perceive the risk of HIV transmission processes, including the source, route, and susceptible population in the MSM population [36,37]. Risk perception values of the source of HIV infection included: Perceived severity of AIDS (very serious, serious, general, not serious, not serious at all), perceived prevalence of HIV among MSM in the participant’s city (very high, high, general, low, very low), and perceived threat of HIV to the participant and their family (very large, relatively large and moderate, very small). Risk perception values of the transmission route included: The frequency of condom use when having anal sex with men (always, occasionally, never). Risk perception values of the susceptible population included the following question: Have you ever engaged in HIV voluntary counseling and testing (VCT)? (no, yes).

### 2.3. Quality Control and Ethics

Based on a large number of studies, this research plan was formulated and demonstrated by experts in infectious diseases, epidemiology, and health statistics. Investigators and quality controllers were trained strictly, and the logicality and integrity of the questionnaire content were checked. This study followed the Helsinki Declaration and was approved by the Ethics Committee of Chongqing Medical University. In addition, the experiment was supervised by the ethics committee, and the participants all signed the informed consent.

### 2.4. Statistical Analysis

The database was established by the Epidata 3.1 software (EpiData Associations, Odense, Denmark), and real-time double entry and logical verification of the data were carried out. Statistical analysis was performed by the SAS 9.4 software (SAS Institute, Cary, NC, USA). The score for anxiety and depression was described by mean ± SD (standard deviation) and the median. Anxiety and depression scores of different anal sex roles were compared by analysis of variance (ANOVA). Univariate analysis of anxiety and depression was performed by the *χ2* test and Fisher’s exact test. The variables were screened using stepwise regression in multivariate logistic regression analysis (sle = 0.05, sls = 0.05), and the odds ratio (OR) value and 95% confidence intervals (CI) were calculated. *p* < 0.05 was considered statistically significant. In addition, the average missing rate of variables was below 0.8%, and observations with missing values were excluded from the analyses (Table 1).

## 3. Results

### 3.1. Participants’ Characteristics

A total of 2422 subjects were recruited from Chongqing, Sichuan, Guangxi, and Xinjiang in western China, and 1771 HIV-negative MSM were eligible (Figure 1). Among them, 467 (26.37%) MSM were classified as anal sex role “1”, 182 (10.28%) as anal sex role “0”, and 1122 (63.35%) as anal sex role “0.5”. The average age of the participants was 29.89 years (median = 28), that of anal sex role “1” was 32.02 years (median = 31), anal sex role “0” was 26.79 years (median = 25), and anal sex role “0.5” was 29.51 years (median = 27). Urban and rural MSM accounted for 71.71% and 28.26%, respectively. The majority of MSM were well educated, of which 61.39% had a college or undergraduate degree or above, 26.46% had a high school or professional high school degree, and only 12.15% had a junior high school or below. A total of 76.81% of MSM had a job, 9.45% were retired or unemployed, and 13.74% were students. Unmarried MSM accounted for 74.20%. Almost half (52.94%) of the participants had an average monthly income of 3000 yuan or less, 34.38% between 3001 and 5000 yuan, and only 12.68% over 5001 yuan (Table 1).

### 3.2. The Prevalence of Anxiety and Depression in Different Anal Sex Roles of MSM

There were 461 (26.03%) HIV-negative MSM who experienced anxiety and 670 (37.83%) HIV-negative MSM who experienced depression in western China (Table 1). The anxiety and depression scores of different anal sex role groups are shown in Table 2. Among all the participants, the mental problems of HIV-negative MSM in anal sex role “0” were the most serious, and the prevalence of anxiety and depression were 31.32% and 46.15%, respectively. The prevalence of anxiety and depression in the anal sex role “0.5” group followed, and that of the anal sex role “1” group was the lowest. Moreover, the *χ*^2^ test suggested that the prevalence of anxiety and depression was significantly different among the anal sex role groups (*p* < 0.05) (Table 2).

### 3.3. The Influencing Factors of Anxiety in HIV-Negative MSM

In the univariate analysis, significant differences in the prevalence of anxiety were observed between groups with different characteristics, including anal sex roles, urban or rural, ethnic groups, educational level, employment status, average monthly income, casual male partners, female partners, frequency of seeking partners on the Internet, commercial sexual service, diagnosed with an STD, alcohol use, perceived threat of HIV, condom use, and HIV counseling (*p* < 0.05) (Table 1).

In the multivariate model (Table 3), anxiety in HIV-negative MSM was associated with anal sex role “0” or “0.5”, low educational level, female sexual partners, frequently seeking sexual partners on the Internet, STD diagnosis, and no HIV counseling (*p* < 0.05). Anal sex role “0” had the highest risk of anxiety, which was 1.783 times higher than anal sex role “1” (OR 1.783, 95% CI 1.188–2.677), and anal sex role “0.5” had a 1.336 times higher risk of anxiety than “1” (OR 1.336, 95% CI 1.021–1.750). Compared with HIV-negative MSM who had a bachelor or graduate degree or above, MSM under junior high school or in senior high school were more likely to experience anxiety (OR 2.276, 95% CI 1.643–3.154; OR 1.366, 95% CI 1.054–1.770, respectively). Furthermore, having female partners in the past six months (OR 1.717, 95% CI 1.285–2.293) or often searching for partners on the Internet (OR 1.793, 95% CI 1.163–2.763) also increased the risk of anxiety. Moreover, MSM who were diagnosed with an STD by a doctor were more likely to suffer from anxiety (OR 1.813, 95% CI 1.242–2.649). However, HIV counseling showed to be a protective factor against anxiety for HIV-negative MSM (OR 0.769, 95% CI 0.613–0.965).

### 3.4. The Influencing Factors of Depression in HIV-Negative MSM

Univariate analysis revealed that the depression of HIV-negative MSM showed a significant difference in anal sex roles, age groups, urban or rural areas, educational level, average monthly income, casual male partners, female partners, frequency of seeking partners on the Internet, diagnosed with an STD, alcohol use, condom use, perceived threat of HIV, and HIV counseling and testing (*p* < 0.05) (Table 1).

Multivariate logistic regression analysis (Table 4) showed that the prevalence of depression was related to anal sex roles “0” and “0.5”, rural area, casual male partners, female partners, diagnosed with an STD, frequently seeking partners on the Internet, no HIV counseling, no condom use, and daily alcohol use (*p* < 0.05). Similar to anxiety, anal sex role “0” and anal sex role “0.5” were exposure factors for depression, and the probability of risk for depression was 1.784 times and 1.309 times that of anal sex role “1” (OR 1.784, 95% CI 1.212–2.625; OR 1.309, 95% CI 1.019–1.682, respectively). Furthermore, the risk of depression in urban HIV-negative MSM was lower than that in rural HIV-negative MSM (OR 0.662, 95% CI 0.525–0.834). In terms of behavior-related factors, MSM with casual male partners (OR 1.319, 95% CI 1.042–1.670) or female partners (OR 1.721, 95% CI: 1.291–2.295) in the past six months increased the appearance of a depression symptom. MSM who were diagnosed with an STD were more likely to be depressed as well (OR 1.673, 95% CI 1.143–2.447). MSM who often sought partners through the Internet in the past six months were 1.947 times more likely to develop depression than those of MSM never finding any online partners (OR 1.947, 95% CI 1.244–3.048). In addition, the model also implied that MSM who drank alcohol every day in the most recent month were more likely to be depressed than MSM who never drank (OR 2.362, 95% CI 1.385–4.029). In regards to risk perception-related factors, compared with MSM without condom use, always or occasional condom use were protective factors for depression in HIV-negative MSM (OR 0.636, 95% CI 0.441–0.918; OR 0.598, 95% CI 0.407–0.879, respectively). Furthermore, the risk of depression in HIV-negative MSM would be significantly decreased when someone had ever received HIV voluntary counseling (OR 0.588, 95% CI 0.473–0.730).

## 4. Discussion

### 4.1. The Prevalence of Anxiety and Depression in HIV-Negative MSM

This study found that mental health problems were common in HIV-negative MSM of western China, and the prevalence of anxiety and depression was not optimistic, especially for MSM having sexual intercourse with men in anal sex role “0” and anal sex role “0.5”.

The prevalence of anxiety in HIV-negative MSM in four western provinces of China was 26.03%, slightly higher than that in Zhejiang Province (24.0%) as the developed area of eastern China [15], and significantly higher than that in Guangzhou City (14.5%) as the developed city of southern China [30]. It suggests that when compared with the developed coastal cities in eastern or southern China, MSM of western China show a rather high prevalence level, perhaps due to a relatively backward economy, a less open culture, and less social support. Furthermore, the prevalence of anxiety in the HIV-negative MSM of our study was lower than that in Chongqing (37.5%) but similar to that in Sichuan (24.0%) in HIV-positive MSM of a previous study. This also indicates that the prevalence of anxiety in HIV-negative MSM is relatively lower than HIV-positive MSM [30] but still at a higher prevalence level. Although few reports have been done on anxiety symptoms in HIV-negative MSM in international studies, studies conducted in Vancouver, Canada [12], and Estonia [26] share similar opinions with ours and have reported a high prevalence as well.

In terms of depression, our study also reported a high prevalence of depression of 37.83%. Compared with other areas in China, our result in the west is in the upper middle level: 22.4% in Shenyang, northeastern China [38]; 34.7% in Guangzhou, southern China [30]; and 45.4% in Zhejiang, eastern China [15]. Moreover, the prevalence varies from area to area, due to the differences in culture, economics, and the development of local mental health services. On account of the relatively lagging development of the four provinces in the west that we engaged, and the grim situation of HIV in MSM in these areas, our study is a better representation of the occurrence of HIV-negative MSM in western China, and converges with internationally reported levels: 36.7% in 1367 MSM in Moscow, Russia [39]; one third in eastern Europe [26]; and 44% in South Africa [40], respectively.

More importantly, our research demonstrated that the mental health condition of HIV-negative MSM varies among the different anal sex roles; however, few studies have compared the anxiety and depression status of MSM with different anal sex roles internationally. Anal sex role “0” of HIV-negative MSM in western China reported the highest risk for anxiety and depression, reaching 31.32% and 46.15%, respectively, which was obviously higher than the average level of MSM [12,14], and even higher than that of anal sex role “0” in Yi’s report [16]. It infers that anxiety and depression symptoms in anal sex role “0” MSM are more serious than other anal sex roles, let alone the general population. Thus, we should pay more attention to mental health and its prevention for anal sex role “0” MSM.

### 4.2. Influencing Factors

In this study, we primarily focused on four factors: Anal sex role factors, social demographic factors, behavior-related factors, and risk perception-related factors. Although numerous studies have demonstrated the contact between demographic factors and mental disorders, we still pointed out that educational level has severely limited the development of mental health, even worse for the mental health of rural MSM. The lack of education restricts access to HIV knowledge and health care services, gradually increasing their psychological burden. Consistent with the findings in the United States [41], Canada [12], and Kunming [14], but contrary to those in Zhejiang rural MSM [15], this study gains a better understanding of MSM in western China, with huge rural areas and a lower level of economy, culture, and health.

Among all the variables in this study, we found that the anal sex role was an important factor affecting both anxiety and depression in HIV-negative MSM. However, most of the existing literature has overlooked differences in mental disorders between different anal sex roles, usually comparing physical problems, such as HIV infection [21]. Multivariate analysis showed that the risk of anxiety and depression in MSM with anal sex role “0” was the highest, which was 1.791 and 1.784 times higher than that of anal sex role “1”, respectively. The possible explanation may be that MSM with anal sex role “0” are usually in a passive and submissive role in male sexual activities, their psychological and physiological needs cannot be fulfilled, and long-term frustrations are prone to anxiety and depression symptoms. Additionally, anal sex role “0” MSM generally are reported to have a younger age [17], first anal sexual intercourse at an early age [42], and frequent unprotected anal sex, which in turn can explain their “submissive” status and weak self-protection ability. Thus, mental health problems are highlighted in HIV-negative MSM with anal sex role “0”. However, MSM with anal sex role “1” are always in the “dominant” position, have more abundant experience, a stronger ability of self-protection, and barely develop mental problems. In line with other research [16], the conflicts of the sex psychology of MSM with anal sex role “0.5” were prominent, caused by their variability of sexual partners and behaviors. Meanwhile, previous studies [17] in China also pointed out that MSM with anal sex role “0” or “0.5” had a higher risk of HIV, perianal Human papillomavirus (HPV), and syphilis infection, which also increased stress and regret after anal intercourse, leading to negative emotions of anxiety or depression. Analogous results have been obtained from MSM research in San Francisco, USA [23] and Lima, Peru [33]. Therefore, more attention should be paid to HIV-negative MSM with an anal sex role of “0” and “0.5” in comprehensive HIV prevention and control strategy in the future.

As the most directly related features of MSM, we found that high-risk sexual behaviors were an important risk factor for anxiety and depression [43]. China is an ancient country with profound ethical traditions. Many Chinese MSM will marry women to cover up their true sexual orientation for the sake of “having a son to carry his family name” or hide their homosexual activities by searching for partners on the Internet or seeking casual sexual partners. Moreover, our study also suggested that we should provide timely treatment and counseling for HIV-negative MSM who have been diagnosed with an STD, so as to reduce the psychological burden and alleviate anxiety and depression. In addition, similar to the report of Chou [12], our results also indicated that HIV-negative MSM who consumed alcohol every day were more likely to suffer from depression, but the causal relationship needs stronger longitudinal research evidence.

It is worth mentioning that our study also found that the higher the risk perception of HIV transmission sources, transmission routes, and susceptible populations, the more likely it is for HIV-negative MSM to develop mental health problems. The risk perception of transmission routes, including condom use in anal sex, was shown to be a protective factor for depression in HIV-negative MSM. Global experience [4] in AIDS prevention and control has proven the effectiveness of condoms in preventing HIV infection. However, in our study, only half of the participants used a condom at any time, up to 10% of which never used a condom. Moreover, studies of 3217 Kenyans [44] and 91,477 Europeans [45] also reported a high percentage of anal intercourse without a condom in MSM. After sex, HIV-negative MSM who did not use condoms would have doubts about their partner’s health and worry about potential HIV infection, and in the long run, depressive symptoms are likely to occur. Studies in eastern Europe [26] and Shenzhen, China [46] also shared the same view with ours that the prevalence of anxiety and depression in HIV-negative MSM who did not use condoms was even higher. In addition, as the highlight of our study, the passive status in sex activities of the anal sex role “0” group restricted the use of condoms, so the proportion of condom use in the “0” group was lower and the risk of anxiety and depression was higher [47,48]. In terms of the risk perception of susceptible groups, MSM with HIV counseling were not likely to experience anxiety and depression, which is consistent with the view of David [26]. It revealed that HIV counseling for MSM could give professional advice to understand their own situation and susceptibility in sexual activities, and help to make appropriate psychological construction and adjustment. In addition, although variables of infection sources of risk perception did not enter the regression model, more propaganda and health education will also be needed to alleviate the “fear of AIDS” in the future.

### 4.3. Limitation

It is important to note several limitations in our study. Notably, due to the particularity of the MSM, the study subjects may have a selection bias by using non-probability sampling methods. Meanwhile, this study was not exactly a randomized trial, and it could be called a real-world study, so there is potential bias of the population. Moreover, all sexual practices were self-reported by the respondents, and authenticity could not be determined. There would be some judgment and potential bias of the practices. Moreover, only cross-sectional data of the baseline in the cohort study were used, so the result obtained could not identify a causal relationship. Thus, longitudinal studies are needed in the future.

## 5. Conclusions

This study found that the prevalence of anxiety and depression in HIV-negative MSM in western China was relatively high, and that of anal sex role “0” MSM was the highest. However, few studies have focused on the mental health of different anal sex roles in HIV-negative MSM, so such differences should be given more attention in the future in China. At the same time, high-risk behaviors and high-risk perceptions also increased the risk of anxiety and depression in HIV-negative MSM. Therefore, the relevant departments of health should provide differentiated health services and carry out early identification and key intervention for high-risk groups with anxiety and depression symptoms. Anxiety and depression symptoms may have an impact on the mental health of these populations in the long run. Furthermore, in this study, MSM sexual practices of anal sex role “0” (receptive anal sex) were associated with mental health problems, like anxiety and depression symptoms, which require more psychological counseling, as well as the understanding and support of the social mainstream. Moreover, this study also suggested that the government should improve the accessibility of HIV counseling and testing and advocate condom use in western China, so that HIV-negative MSM with low educational levels and MSM in rural areas can also get timely and better health and medical care for anxious and depressive emotions. In addition, previous studies [49,50,51] also found that the prevalence of anxiety and depression in HIV-negative MSM affects adherence to HIV prevention and treatment, so early identification of anxiety and depression symptoms and exploration of influencing factors in HIV-negative MSM conducted in our study shows particular importance for the mental health of MSM in western China.

## Figures and Tables

**Figure 1 ijerph-17-00464-f001:**
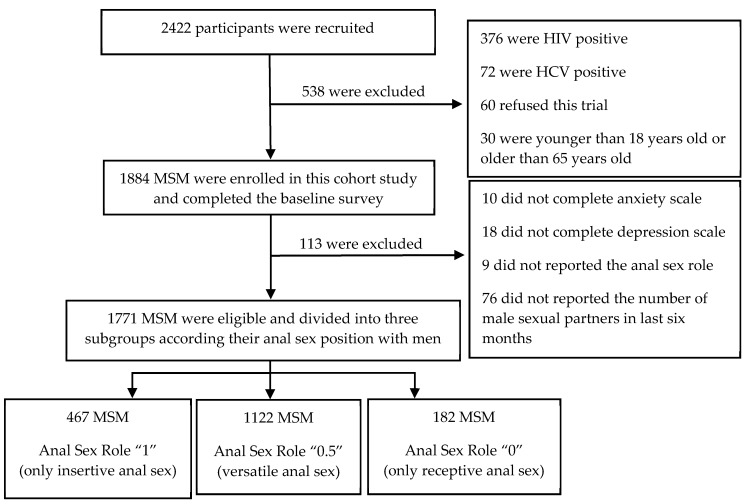
Flow chart of participants’ enrollment.

**Table 1 ijerph-17-00464-t001:** Sample characteristics of human immunodeficiency virus negative (HIV-negative) men who have sex with men (MSM) and univariate analysis.

Characteristics	Total *n* (%)	Anxiety		*p* Value	Depression		*p* Value
		NO *n* (%)	YES *n* (%)		NO *n* (%)	YES *n* (%)	
***N***	**1771 (100.00)**	**1310 (73.97)**	461 (26.03)		1101 (62.17)	670 (37.83)	
**Anal sex role factors**							
The role of MSM in anal sex with male sexual partners							
Anal sex role “1”	467 (26.37)	363 (77.73)	104 (22.27)	**0.0415**	314 (67.24)	153 (32.76)	**0.0047**
Anal sex role “0.5”	1122 (63.35)	822 (73.26)	300 (26.74)		689 (61.41)	433 (38.59)	
Anal sex role “0”	182 (10.28)	125 (68.68)	57 (31.32)		98 (53.85)	84 (46.15)	
**Social demographic factors**							
Age							
18–24	557 (31.45)	400 (71.81)	157 (28.19)	0.1633	321 (57.63)	236 (42.37)	**0.0285**
25–30	532 (30.04)	389 (73.12)	143 (26.88)		342 (64.29)	190 (35.71)	
≥31	682 (38.51)	521 (76.39)	161 (23.61)		438 (64.22)	244 (35.78)	
Area ^#^							
Urban	1267 (71.74)	971 (76.64)	296 (23.26)	**<0.0001**	829 (65.43)	438 (34.57)	**<0.0001**
Rural	499 (28.26)	337 (67.54)	162 (32.46)		271 (54.31)	228 (45.69)	
Ethnic groups ^#^							
Han nationality	1638 (92.54)	1201 (73.32)	437 (26.68)	**0.0324**	1009 (61.60)	629 (38.40)	0.0944
Other ethnic minorities	132 (7.46)	108 (81.82)	24 (18.18)		91 (68.94)	41 (31.06)	
Educational level ^#^							
Junior high or below	215 (12.15)	128 (59.53)	87 (40.47)	**<0.0001**	108 (50.23)	107 (49.77)	**<0.0001**
Senior high	468 (26.46)	330 (70.51)	138 (29.49)		275 (58.76)	193 (41.24)	
College or graduate or above	1086 (61.39)	850 (78.27)	236 (21.73)		716 (65.93)	370 (34.07)	
Employment status ^#^							
On the job	1358 (76.81)	1016 (74.82)	342 (25.18)	**0.0150**	861 (63.40)	497 (36.60)	0.1219
Students at school	243 (13.74)	184 (75.72)	59 (24.28)		144 (59.26)	99 (40.74)	
Retired or unemployed	167 (9.45)	108 (64.67)	59 (35.33)		94 (56.29)	73 (43.71)	
Marital status							
Unmarried	1314 (74.20)	986 (75.04)	328 (24.96)	0.0706	834 (63.47)	480 (36.53)	0.1257
Married	307 (17.33)	211 (68.73)	96 (31.27)		176 (57.33)	131 (42.67)	
Divorce or widowhood	150 (8.47)	113 (75.33)	37 (24.67)		91 (60.67)	59 (39.33)	
Personal monthly income ^#^							
≤3000	927 (52.94)	658 (70.98)	269 (29.02)	**0.0014**	522 (56.31)	405 (43.69)	**<0.0001**
3001–5000	602 (34.38)	454 (75.42)	148 (24.58)		398 (66.11)	204 (33.89)	
≥5001	222 (12.68)	183 (82.43)	39 (17.57)		168 (75.68)	54 (24.32)	
**Behavior-related factors**							
Regular male sexual partners in the last six months ^#^							
0	293 (16.80)	217 (74.06)	76 (25.94)	0.2290	179 (61.09)	114 (38.91)	0.3778
1	1188 (68.12)	892 (75.08)	296 (24.92)		755 (63.55)	433 (36.45)	
≥2	263 (15.08)	184 (69.96)	79 (30.04)		156 (59.32)	107 (40.68)	
Casual male sexual partners in the last six months ^#^							
0	705 (40.68)	541 (76.74)	164 (23.26)	**0.0373**	466 (66.10)	239 (33.90)	**0.0072**
≥1	1028 (59.32)	743 (72.28)	285 (27.72)		614 (59.73)	414 (40.27)	
Female sexual partners in the last six months ^#^							
0	1462 (84.17)	1113 (76.13)	349 (23.87)	**<0.0001**	944 (64.57)	518 (35.43)	**<0.0001**
≥1	275 (15.83)	172 (62.55)	103 (37.45)		137 (49.82)	138 (50.18)	
Frequency of seeking partners on the Internet in the last 6 months ^#^							
Often	117 (6.62)	69 (58.97)	48 (41.03)	**0.0006**	55 (47.01)	62 (52.99)	**0.0008**
Sometimes or occasionally	986 (55.80)	743 (75.35)	243 (24.65)		612 (62.07)	374 (37.93)	
Never	664 (37.58)	495 (74.55)	169 (25.45)		434 (65.36)	230 (34.64)	
Whether a commercial sexual service in the past 6 months ^#^							
Yes	100 (5.66)	63 (63.00)	37 (37.00)	**0.0095**	54 (54.00)	46 (46.00)	0.0810
No	1668 (94.34)	1246 (74.70)	422 (25.30)		1046 (62.71)	622 (37.29)	
Diagnosed with an sexually transmitted diseases (STD) in the past six months ^#^							
Yes	142 (8.03)	87 (61.27)	55 (38.73)	**0.0003**	70 (49.30)	72 (50.70)	**0.0009**
No	1626 (91.97)	1221 (75.09)	405 (24.91)		1030 (63.25)	596 (36.65)	
Frequency of alcohol use in the last month ^#^							
Daily	72 (4.07)	45 (62.50)	27 (37.50)	**0.0464**	30 (41.67)	42 (58.33)	**0.0009**
Occasionally	1152 (65.12)	866 (75.17)	286 (24.83)		733 (63.63)	419 (36.37)	
Never	545 (30.81)	397 (72.84)	148 (27.16)		336 (61.65)	209 (38.35)	
Whether a use of illicit drugs in the past six months ^#^							
No	1695 (96.91)	1261 (74.40)	434 (25.60)	0.1136	1058 (62.42)	637 (37.58)	0.8452
Yes	54 (3.09)	35 (64.81)	19 (35.19)		33 (61.11)	21 (38.89)	
**Risk perception-related factors**							
Perceived severity of AIDS ^#^							
Very serious	1107 (62.54)	832 (75.16)	275 (24.84)	0.3072 *	679 (61.34)	428 (38.66)	0.5051 *
Serious	522 (29.49)	379 (72.61)	143 (27.39)		334 (63.98)	188 (36.02)	
General	123 (6.95)	87 (70.73)	36 (29.27)		77 (62.60)	46 (37.40)	
Not serious	10 (0.56)	7 (70.00)	3 (30.00)		7 (70.00)	3 (30.00)	
Not serious at all	8 (0.45)	4 (50.00)	4 (50.00)		3 (37.50)	5 (62.50)	
Perceived prevalence of HIV among MSM in your city ^#^							
Very high	375 (21.23)	258 (68.80)	117 (31.20)	0.0691	219 (58.40)	156 (41.60)	0.2287
High	774 (43.83)	595 (76.87)	179 (23.13)		502 (64.86)	272 (35.14)	
General	420 (23.78)	308 (73.33)	112 (26.67)		253 (60.24)	167 (39.76)	
Low	126 (7.13)	93 (73.81)	33 (26.19)		81 (64.29)	45 (35.71)	
Very low	71 (4.02)	52 (73.24)	19 (26.76)		43 (60.56)	28 (39.44)	
Perceived threat of HIV to yourself and your family ^#^							
Very large	973 (55.03)	707 (72.66)	266 (27.34)	**0.0994**	587 (60.33)	386 (39.67)	**0.0147**
Relatively large and moderate	639 (36.14)	474 (74.18)	165 (25.82)		400 (62.60)	239 (37.40)	
Very small	156 (8.82)	126 (80.77)	30 (19.23)		113 (72.44)	43 (27.56)	
Frequency of condom use when having anal sex with men ^#^							
Always	956 (56.84)	725 (75.84)	231 (24.16)	**0.0051**	616 (64.44)	340 (35.56)	**0.0050**
Occasionally	562 (33.41)	423 (75.27)	139 (24.73)		356 (63.35)	206 (36.65)	
Never	164 (9.75)	105 (64.02)	59 (35.98)		84 (51.22)	80 (48.78)	
HIV voluntary counseling ^#^							
Yes	1074 (60.88)	820 (76.35)	254 (23.65)	**0.0046**	719 (66.95)	355 (33.05)	**<0.0001**
No	690 (39.12)	485 (70.29)	205 (29.71)		377 (54.64)	313 (45.36)	
HIV voluntary testing ^#^							
Yes	1362 (77.17)	1021 (74.96)	341 (25.04)	0.1586	870 (63.88)	492 (36.12)	**0.0133**
No	403 (22.83)	288 (71.46)	115 (28.54)		230 (57.07)	173 (42.93)	

* Fisher’s exact test and *χ*^2^ test was used for the rest. ^#^ indicates loss of data. Bold values indicate statistical significance at *p* < 0.05.

**Table 2 ijerph-17-00464-t002:** Scores and prevalence of anxiety and depression in different anal sex roles.

Mental Health Problems	Anal Sex Role “1” (*n* = 467)	Anal Sex Role “0.5” (*n* = 1122)	Anal Sex Role “0” (*n* = 182)	*F/χ* ^2^	*p*
	Score/*N* (%)	Score/*N* (%)	Score/*N* (%)		
**Anxiety**					
Mean ± SD	42.09 ± 10.20	42.94 ± 10.75	44.25 ± 11.55	2.80	0.0611 ^a^
Median	42.00	42.00	42.00		
Yes (≥50)	104 (22.27)	300 (26.74)	57 (31.32)	6.3652	**0.0415 ^b^**
No (<50)	363 (77.73)	822 (73.26)	125 (68.68)		
**Depression**					
Mean ± SD	12.78 ± 8.93	14.02 ± 9.62	16.31 ± 11.71	8.83	**0.0002 ^a^**
Median	12.00	12.00	13.50		
Yes (≥16)	153 (32.76)	433 (38.59)	84 (46.15)	10.7378	**0.0047 ^b^**
No (<16)	314 (67.24)	689 (61.41)	98 (53.85)		

^a^ Analysis of variance. ^b^
*χ*^2^ test. Bold values indicate statistical significance at *p* < 0.05.

**Table 3 ijerph-17-00464-t003:** Multivariate logistic stepwise regression of anxiety in HIV-negative MSM.

Independent Variables	B	S.E.	Wald	*p* Value	OR	95%CI for OR	
						Lower	Upper
Anal sex role							
“1”							**(reference)**
“0.5”	0.2900	0.1374	4.4534	**0.0348**	1.336	1.021	1.750
“0”	0.5783	0.2073	7.7802	**0.0053**	1.783	1.188	2.677
Educational level							
College or graduate or above							**(reference)**
Senior high	0.3117	0.1322	5.5604	**0.0184**	1.366	1.054	1.770
Junior high or below	0.8225	0.1664	24.4415	**<0.0001**	2.276	1.643	3.154
Female sexual partners in the last six months							
0							**(reference)**
≥1	0.5404	0.1478	13.3661	**0.0003**	1.717	1.285	2.293
Frequency of seeking partners on the Internet in the last 6 months							
Never							**(reference)**
Sometimes or occasionally	−0.0636	0.1221	0.2715	0.6023	0.938	0.739	1.192
Often	0.5836	0.2207	6.9947	**0.0082**	1.793	1.163	2.763
Diagnosed with STD in the past six months							
No							**(reference)**
Yes	0.5953	0.1933	9.4873	**0.0021**	1.813	1.242	2.649
HIV voluntary counseling							
No							**(reference)**
Yes	−0.2628	0.1159	5.1445	**0.0233**	0.769	0.613	0.965

Dependent variable = anxiety. Bold values indicate statistical significance at *p* < 0.05. Statistically significant variables (*p* < 0.05) in univariate analysis were included in the model, and variables with multiple collinearity with anal sex role were excluded: Age, marital status, and personal monthly income.

**Table 4 ijerph-17-00464-t004:** Multivariate logistic stepwise regression of depression in HIV-negative MSM.

Independent Variables	B	S.E.	Wald	*p* Value	OR	95%CI for OR	
						Lower	Upper
Anal sex role							
“1”							**(reference)**
“0.5”	0.2693	0.1278	4.4395	**0.0351**	1.309	1.019	1.682
“0”	0.5787	0.1971	8.6243	**0.0033**	1.784	1.212	2.625
Area							
Rural							**(reference)**
Urban	−0.4127	0.1182	12.1885	**0.0005**	0.662	0.525	0.834
Casual sexual partners in the last six months							
0							**(reference)**
≥1	0.2772	0.1202	5.3133	**0.0212**	1.319	1.042	1.670
Female sexual partners in the last six months							
0							**(reference)**
≥1	0.5432	0.1467	13.7131	**0.0002**	1.721	1.291	2.295
Diagnosed with STD in the past six months							
No							**(reference)**
Yes	0.5144	0.1941	7.0253	**0.0080**	1.673	1.143	2.447
Frequency of seeking partners on the Internet in the last 6 months							
Never							**(reference)**
Sometimes or occasionally	0.0124	0.1233	0.0101	0.9198	1.012	0.795	1.289
Often	0.6664	0.2287	8.4909	**0.0036**	1.947	1.244	3.048
Frequency of alcohol use in the last month							
Never							**(reference)**
Occasionally	−0.1704	0.1177	2.0954	0.1477	0.843	0.670	1.062
Daily	0.8594	0.2725	9.9499	**0.0016**	2.362	1.385	4.029
Frequency of condom use when having anal sex with men							
Never							**(reference)**
Occasionally	−0.5143	0.1966	6.8456	**0.0089**	0.598	0.407	0.879
Always	−0.4524	0.1871	5.8425	**0.0156**	0.636	0.441	0.918
HIV voluntary counseling							
No							**(reference)**
Yes	−0.5317	0.1104	23.1937	**<0.0001**	0.588	0.473	0.730

Dependent variable = depression. Bold values indicate statistical significance at *p* < 0.05. Statistically significant variables (*p* < 0.05) in univariate analysis were included in the model, and variables with multiple collinearity with anal sex role were excluded: Age, marital status, and personal monthly income.

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
