# Peer review of "Anxiety and Depression Associated with Anal Sexual Practices among HIV-Negative Men Who Have Sex with Men in Western China"

_ijerph, 2020, doi:10.3390/ijerph17020464_

Round 1
Reviewer 1 Report
Comments:
This article described the prevalence and factors of anxiety and depression differentiated by anal sex roles among HIV-negative men who have sex with men in western China, which is interesting. But the language needs to be improved a lot and also need to think more about the public health implications. The topic in interesting, but the writing needs to be improved a lot.
Please find a native English speaker to do the proofreading before resubmission.
Introduction
Line31, “the prevalence of AIDS” it’s a bit odd. Normally, we say the HIV prevalence.
Line 33, “cause of their special sexal behaviors”, this is not quite coherent
Materials and methods
Line 128-129, 2.2.5 we normally don’t say fixed male sexual partner or temporary, we say “regular” or “casual” sexual partners
Line 136-144, 2.2.6 the definition of risk perception, you mentioned “perceived severity, perceived prevalence…, and perceived threat…..” is this from Health Belief Model, you had a reference of Dr. Ma’s epidemic model, but I am not sure is that commonly used?
In Figure 1, the left box, “10 did not complete the anxiety scale”, “108 did not complete the anxiety scale”, a typo here?
Results
Line 188, “Moreover, ANOVA found that there was no significant difference in anxiety scores between different anal sex roles in HN MSM’. But in Table 1, there is “0.0415”, it seems significant, correct??
In Table 1, You used comorbidity of anxiety and depression, why you suddenly introduced the “comorbidity” term? Comorbidity will normally when an additional chronic condition occurs to someone who has an existing disease. For instance, if you investigate HIV-positive MSM, you can say they comorbid with depression or anxiety. Here the term is not appropriate, consider to change.
Discussion
Line 258-259 I don’t think it’s a good idea to use HP and HN for HIV-positive and HIV-negative, it is confusing sometimes. May consider just spell them out.
Reviewer 2 Report
This article is interesting research done in Western China. However, the topic itself is not a novel idea and will have low interest in readership of this journal. Previous similar research has been conducted on this topic in China. The title is too long and could capture in tis title general concepts like mental well-being/health, sexual practices among men who have sex with men, for example, in parlance with the current international literature. The words are too narrow and specific in the title itself.
There appears to be an underlying philosophy or thought bordering on judgment in that references are made to mental disorders and related sexual practices. See 319-320 and 318-319 references to weakness of self-restraint; reference to bad emotions. An assumption is made that men who engage in these practices need alleviation from mental disorders. This is alluded to right in the beginning of the abstract.This is dated and attitudes have shifted from viewing this topic from a pathological perspective to one where behaviors have been "mainstreamed" and understood without the stereotypical "illness model". The DSM -5 and psychiatric literature do not uphold this perspective. Maybe just refer to depression and anxiety? And follow that throughout the manuscript as central foci?
The descriptions of and categories of the type pf anal sex are cumbersome and awkward to follow and understand. 117 2.2.3 Anal sex role factors the description is not clear and confusing. As a result, it proves difficult for the reader to make sense out of the interpretation of the results:167-170.Ditto for Figure 1
The manuscript could benefit from serious editing. The greatest limitation here is not what the authors list at the end of the manuscript but the limitations in clarity, judgement, and potential bias of the population and the practices. A description of psychological factors challenging mental health like anxiety and depression, would have been more palatable and tying that to what subjects revealed in relation to their sexual practices, would have been more informative and instructive.
Round 2
Reviewer 1 Report
The first sentence of the introduction: negative trend? Quite confusing, do you mean "the HIV prevalence among key population is rising?"
The author used "reseaches", i am not sure if the plural form of reseach is researches or still just research?
Author Response
Response to Reviewer 1 Comments
Dear Editors and Reviewers:
Thank you for your letter and for the reviewers’ comments concerning our manuscript (ID: ijerph-664713). We have studied comments carefully and have made correction which we hope meet with approval. The main corrections in the revised paper and the responds to the reviewer’s comments are as follows:
1. The first sentence of the introduction: negative trend? Quite confusing, do you mean "the HIV prevalence among key population is rising?"
Response to comment 1:
Thank you for your kind advice, we have revised the statement as “ In China, the HIV prevalence has declined in recent years, but that among the key population isn’t encouraging which is on the rise. ” (highlighted in red color in the revised paper)
2. The author used "reseaches", i am not sure if the plural form of reseach is researches or still just research?
Response to comment 1:
Sorry for our error, we have unified the "researches" in the full text as "research".
Once again, thank you very much for your warm work and your valuable comments and suggestions.
Sincerely